# Natural Stones with a Self-Cleaning Surface via Self-Assembled Monolayers

Zhuoqi Duan [1], Zaixin Xie [1], Bao Zhou [1], Xiaobo Yang [1], Heng-Yong Nie [2] and Yongmao Hu [1,*]

1   College of Engineering, Dali University, Dali 671003, China; zhuoqiduan@126.com (Z.D.); yndlxzx@126.com (Z.X.); bzhou3@163.com (B.Z.); yangxb@dali.edu.cn (X.Y.)
2   Surface Science Western, The University of Western Ontario, London, ON N6A 3K7, Canada; hnie@uwo.ca
*   Correspondence: yongmaohu@163.com

**Abstract:** Heritage buildings and monuments are mostly made from natural stone, which undergoes irreversible decay under outdoor conditions. The main reason for the contamination, degradation, and cracking of natural stones is water and oil permeation. Hence, modifications on stones rendering their surface self-cleaning are effective for stone protection. Reported in this paper is the development of a bionic approach to enabling self-cleaning stone surface via growing self-assembled polydopamine (PDA) as the adhesive layer on the stone surface, followed by depositing $Al_2O_3$ nanoparticles derivatized by self-assembled monolayers of a fluorophosphonic acid (FPA). This approach ensured a robust surface modification that realized superhydrophobicity, as demonstrated on natural marbles, Hedishi, and Qingshi. The surface modification was thermally stable up to 400 °C.

**Keywords:** self-cleaning stones; surface modification; polydopamine coatings; fluorophosphonic acid (FPA); self-assembled monolayers (SAMs); SAM-derivatized alumina particles; micro- and nanostructures





## 1. Introduction

Heritage buildings and monuments are mostly made from natural stones because of their earth-abundance, bio-friendliness, and robustness. However, natural stones undergo weathering and dissolution, especially when they are exposed to rains containing $SO_2$ and $NO_x$, which are common nowadays. Contaminants invade the stone bodies, along with water, which causes serious degradation to the stones. Hence, surface modifications aiming at increased water repellency are an important route toward stone protection [1–4].

Coatings of acrylic, paraffin, acryl-siliconic, and epoxy resin have been generally used for stone protection [5–11]. Although the pollution sources can be effectively isolated by these coatings, the breathability (i.e., the moisture permeability) of coated stones might be seriously weakened, especially when the coating is thick. As a consequence, depending on the variation of the ambient temperature, the moisture trapped inside a coated stone experiences cycles of crystallization–dissolution, and such cycles accelerate the degradation of the coated stone. Especially in heavily salt-laden cases, simple treatment by seal coatings can even increase the deterioration fate of treated stones, leading to irreversible scaling, flaking, and cracking [12]. Moreover, the stability of these coatings is limited due to their relatively weak adhesion to the stone. Hence, for a protective coating to function properly on stones, its water repellence, robust adhesion to the stone, and breathability are required.

For natural stone protection, self-cleaning surfaces are effective due to their excellent water repellency and breathability [13,14]. Self-cleaning surfaces require low surface energy and a morphology having roughness at both the micro- and nanoscale [15,16]. Intrinsic micro-roughness exists on natural stone surfaces, while the addition of nanoparticles in protectants effectively produces nano-roughness. For example, a static water contact angle (CA) of 160° was shown on marble treated with polyalkylsiloxane mixed with silica nanoparticles [17]. Another protectant was a copolymer of epoxy and acrylate, with an additive of nano $CaCO_3$ nanoparticles [18,19]. The performances of durability

and resistance to abrasion, contamination, and weathering of natural stones have been greatly improved via the application of those protectants. Meanwhile, the antibacterial and antifungal properties of the treated stones were enhanced, as well. The main issue found in the nanomaterials-enhanced protectants is their adhesion to the stone substrate. Although there are numerous polymers that can be used as binders, they hardly possess the desired properties of adhesion and water repellency. In addition, dispersions of nanoparticles in these polymers are generally poor.

Reported in this work is a new approach to achieving self-cleaning stone surfaces that includes a self-assembled layer of polydopamine (PDA) on the stone surface to serve as the adhesive layer and $Al_2O_3$ nanoparticles derivatized with self-assembled monolayers (SAMs) of a fluorophosphonic acid (FPA) to be deposited on the PDA layer. Using natural marble, Hedishi, and Qingshi, we demonstrated that the stone substrate and the FPA derivatized $Al_2O_3$ particles were strongly glued together, due to the robust adhesiveness of PDA layers, resulting in excellent water repellency, as evidenced by highly water static CAs. The thermal stability of the self-cleaning stone surfaces was also investigated.

## 2. Materials and Methods

### 2.1. Materials

Natural marble, Hedishi, and Qingshi blocks were obtained from the local area of Dali prefecture, Yunnan province, China. The dominant ingredient of marble is $CaCO_3$, while that of Hedishi and Qingshi is $SiO_2$. In addition, small amounts of ferric oxides and graphite exist in the stones of Hedishi and Qingshi [20]. $Al_2O_3$ (99.9%, 30 nm) nano-powder was obtained from Shanghai Macklin Biochemical Technology Co., Ltd. (Shanghai, China). Dopamine hydrochloride (98%) and tris-HCl buffer solution (1 M/L, pH = 9) were purchased from Shanghai Aladdin Bio-Chem Technology Co., Ltd. (Shanghai, China), and Beijing Solarbio Science & Technology Co., Ltd. (Beijing, China), respectively. The FPA, $C_{10}F_{21}C_2H_4PO(OH)_2$, was ordered from Specific Polymers (Castries, France). All chemicals were of analytical grade and used without further purification.

### 2.2. Methods

Natural stone blocks were cut into 50 mm × 15 mm × 5 mm cuboid coupons. Both upper and lower surfaces of each coupon were polished by 400# grinding wheels primarily and glued on glass slides by a UV-curable adhesive. The upper surfaces of natural stone specimens were sequentially polished by 1200# grinding wheels and a polishing cloth, followed by a 15 min of ultrasonication in ethanol and deionized (DI) water (18.2 MΩ), successively, to clean the surface. The specimens were dried under 40 °C in a drying box for 24 h, followed by a 3 min UV–ozone cleaning prior to surface modifications toward water repellency.

The Tris-HCl buffer solution was diluted by DI water to 0.01 M/L, and its pH value was adjusted to 8.5 by slowly adding HCl solution. Cleaned stone specimens were immersed into the dopamine hydrochloride Tris-HCl solution in a dark environment for 24 h to polymerize PDA layers on the stone surfaces. After the polymerization was performed, the stone specimens were rinsed with DI water, followed by spinning (3000 rad/s for 30 s) to remove excessive (loose) substance from the polymerized dopamine surface. The specimens were stored in a drying box at 40 °C for 24 h.

The $Al_2O_3$ and FPA powder was heated at 100 °C on a heating plate for 1 h to eliminate moisture prior to use. Colloidal dispersions of $Al_2O_3$@FPA were prepared as follows. $Al_2O_3$ and FPA powders were separately ultrasonicated in 15 mL of ethanol. The $Al_2O_3$ in ethanol emulsion was stirred 3 h to fully disperse the particles. The $Al_2O_3$ emulsion and FPA solution were slowly mixed to obtain $Al_2O_3$ derivatized by self-assembled monolayers of FPA, denoted as $Al_2O_3$@FPA. In the stage of the self-cleaning stone surfaces' formation, the PDA-layer-coated stone specimens were immersed into the $Al_2O_3$@FPA dispersion for 24 h, followed by rinsing with ethanol and drying at 40 °C for 24 h. The sample preparation processes are illustrated in Figure 1.

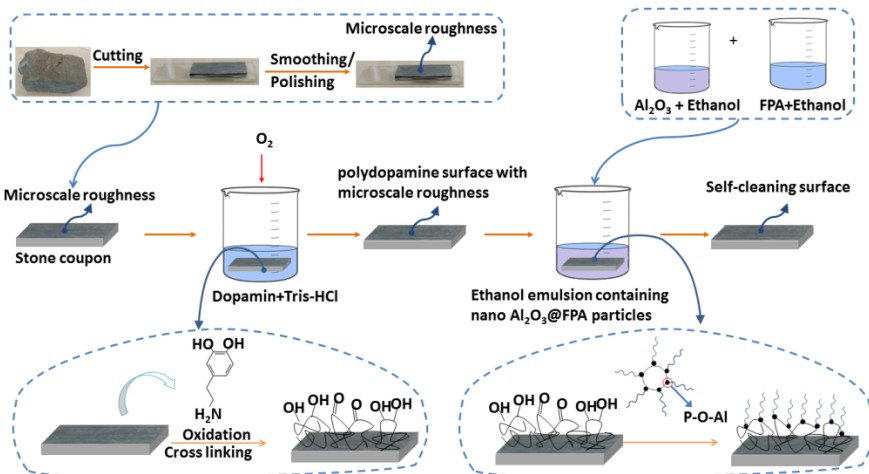

**Figure 1.** Schematic illustration of the processes to render self-cleaning stone surface via formation of polymerized dopamine layer of the stone surface and coating of $Al_2O_3$ particles derivatized with self-assembled monolayers of a fluorophosphonic acid (denoted as $Al_2O_3$@FPA).

### 2.3. Characterizations

Scanning electron microscopy (SEM), SU8020 (Hitachi), was used to investigate the surface of the specimen sputter-coated with a thin gold film for charge compensation. The base pressure of the SEM was $10^{-5}$ Torr, and the instrument was operated at an accelerating voltage of 3.0 kV and a working distance of 3.2 mm. Water CAs were measured with an SDC-200 CA measurement system. All CAs were obtained with DI water or peanut oil as the probing liquid at room temperature, i.e., 23–25 °C, and a relative humidity of 60–75%. The volumes of the droplets used were 3 μL (for water) and 5 μL (for peanut oil), respectively. On each surface, CAs were measured at least on 5 spots, and the averaged values were adopted [21,22].

In order to evaluate the thermal stability of the as-prepared self-cleaning surfaces, marble, Hedishi, and Qingshi surfaces were annealed in air at 100, 150, 200, 250, 300, 350, 400, and 450 °C for 30 min on a hot plate, respectively. After being naturally cooled to room temperature, the surfaces were subjected to water CA measurement. The maximum thermally stable temperature for the self-cleaning surface was defined as the annealing temperature after which an obvious decline of the water CA to below 90° was observed.

## 3. Results

### 3.1. Water Wettability of PDA

In order to demonstrate the formation of PDA layers and to evaluate the water wettability of PDA surfaces. Static water CAs were measured on blank and dopamine Tris-HCl solution–treated glass and stone surfaces. The results were summarized in Table 1. All blank surfaces are hydrophilic, with static water CAs ranging from 45.3° to 57.8°. When coated with PDA layers, the CAs dramatically decreased due to the catechol groups of dopamine. The CAs remained largely unchanged with the various dopamine concentrations.

**Table 1.** Variation of static water CAs on blank and PDA-covered glass and stone surfaces as a function of dopamine concentration in Tris-HCl solution.

| Dopamine Concentration (mg/mL) | Static Water CA (°) | | | |
|---|---|---|---|---|
| | Glass | Marble | Hedishi | Qingshi |
| 0 | 57.8 ± 2.3 | 52.8 ± 1.9 | 45.3 ± 0.8 | 48.1 ± 1.5 |
| 0.5 | 5.9 ± 0.9 | 10.9 ± 1.6 | 11.5 ± 0.9 | 13.2 ± 0.7 |
| 1.0 | 4.1 ± 0.8 | 10.0 ± 0.6 | 12.0 ± 0.9 | 13.2 ± 1.3 |
| 1.5 | 3.1 ± 0.1 | 11.1 ± 1.3 | 12.0 ± 1.1 | 12.6 ± 0.7 |

### 3.2. Impact of PDA Layers on Water/Oil Wettability of the Stone Surfaces

In order to investigate the impact of the PDA layers on the liquid wettability of the stone surfaces, dopamine Tris-HCl solution with concentrations of 0.75, 1.5, 2, 5, and 8 mg/mL were adopted in the polymerization, while the concentration of $Al_2O_3$@FPA was fixed at 1.0 mg/mL. Static, advancing, and receding water and peanut-oil CAs on the treated stones are shown in Figure 2.

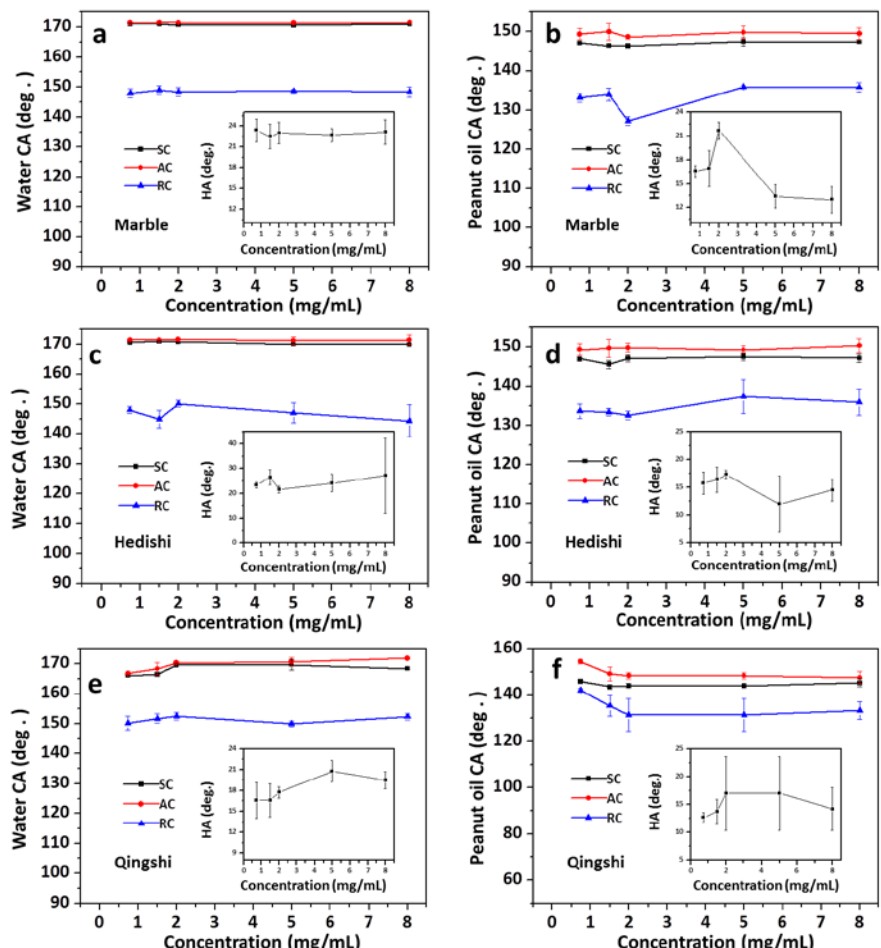

**Figure 2.** Static, advancing, and receding contact angles of water (represented as SC, AC, and RC, respectively) (**a**,**c**,**e**) and peanut oil (**b**,**d**,**f**) on marble, Hedishi, and Qingshi as a function concentration of dopamine polymerized on the stones as the adhesive layer, on which the concentration of $Al_2O_3$@FPA used to render the surface hydrophobic had a concentration 1.0 mg/mL, respectively. Inserted in the figures are the hysteric angles (HAs), that is, the difference between the advancing and receding contact angles.

As shown in Figure 2a,c, the static and advancing water CAs hardly show differences and are in the range of 165–170° for the treated marble and Hedishi for all tested dopamine concentrations. On the other hand, as shown in Figure 2e, for the treated Qingshi surface, when the dopamine concentration is 1.5 mg/mL, its water CAs are smaller and remain unchanged when the concentration is larger than 2 mg/mL. Thus, the treated stones are superhydrophobic, which is necessary for self-cleaning surface.

As shown in Figure 2b,d,f, the advancing CAs of peanut oil on the three treated stones are around 150°, and the static counterparts are slightly lower, reaching approximately 145°. This experimental result suggests that the treated surface is not only superhydrophobic but also superoleophobic.

As shown in the inserts in Figure 2, the water HAs, or the difference between the advancing and receding contact angles, are in the range of 20–25°, while the peanut-oil HAs are in the range of 10–20°.

Figure 3 shows SEM images of a PDA layer polymerized on a silicon wafer surface with a dopamine of 2 mg/mL for the polymerization. Spherical aggregations with a diameter of about 200 nm are observed on a continuous and dense PDA film with a thickness of ~1 μm. Hence, the reason for the slight variation of the water and oil CAs with the concentration of dopamine comes from the smaller roughness of PDA surfaces compared with the intrinsic micro-roughness of the stone surfaces and the micro- and nano-roughness formed by $Al_2O_3$@FPA particles. Since 2 mg/mL dopamine is enough for PDA layer formation and for self-cleaning surfaces construction, in the following experiments, the concentration of dopamine was fixed at 2 mg/mL.

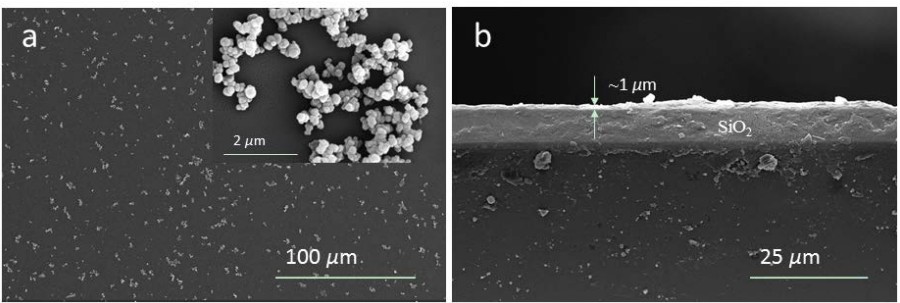

**Figure 3.** SEM images of a PDA layer formed on a silicon wafer surface, showing aggregates on the surface (**a**) and a PDA layer thickness of ~1 μm (**b**). The concentration of dopamine used for the polymerization is 2 mg/mL.

SEM images of the surfaces and cross-sections of three blank stones are shown in Figure 4. On the polished stone surfaces, nonuniform interstices and holes are distinct. The width of the interstices and the diameter of the holes are in the range of several hundred nanometers, and the length of the interstices is mainly several micrometers (Figure 4a–c). From the cross-sections of the three stones shown in Figure 4d–f, we see that the depth of these interstices and holes ranges from nanometers to micrometers.

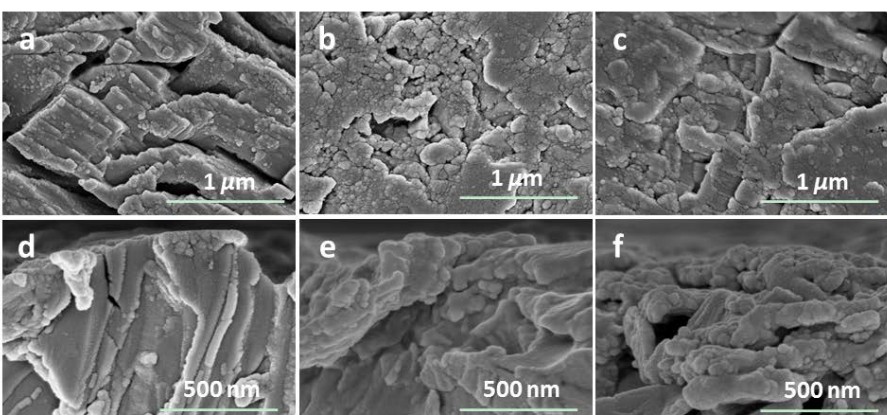

**Figure 4.** SEM images of the surface and cross section of a blank marble (**a**,**d**), Hedishi (**b**,**e**), and Qingshi (**c**,**f**), respectively.

SEM images of the surfaces and cross-sections of three stones after being treated with PDA and $Al_2O_3$@FPA are shown in Figure 5. The stone surfaces present hierarchical structures, including micro- and nanoscale roughness. As shown in Figure 5d–f, the PDA and $Al_2O_3$@FPA particles were permeated into the interstices and holes of the stones beneath the surfaces, and protective layers were formed on the inner surfaces of the interstices and holes, as well.

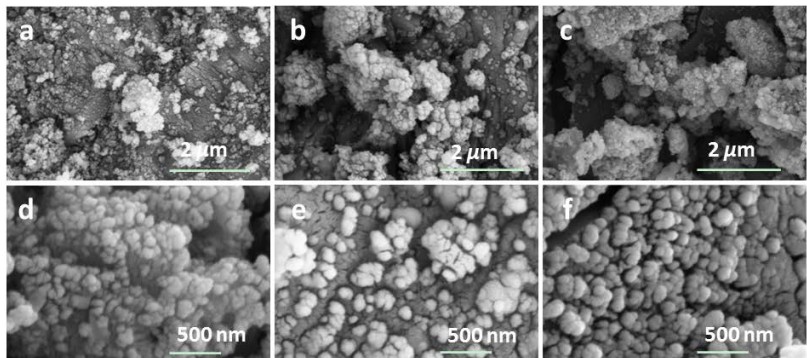

**Figure 5.** SEM images of the surface of marble (**a**), Hedishi (**b**), and Qingshi (**c**) treated with PDA and Al$_2$O$_3$@FPA nanoparticles. Their cross-section SEM images are shown in (**d**–**f**), respectively.

### 3.3. Thermal Stability and Durability

Figure 6a–c shows the static, advancing, and receding water CAs of the control and the water-repellant surfaces prepared on the three stones as a function of annealing temperature. The CAs of the three treated stones remained constant (~170°) after they were annealed at temperatures up to 400 °C. From Figure 6d–f, we see that the micro- and nanostructures were well maintained during the annealing.

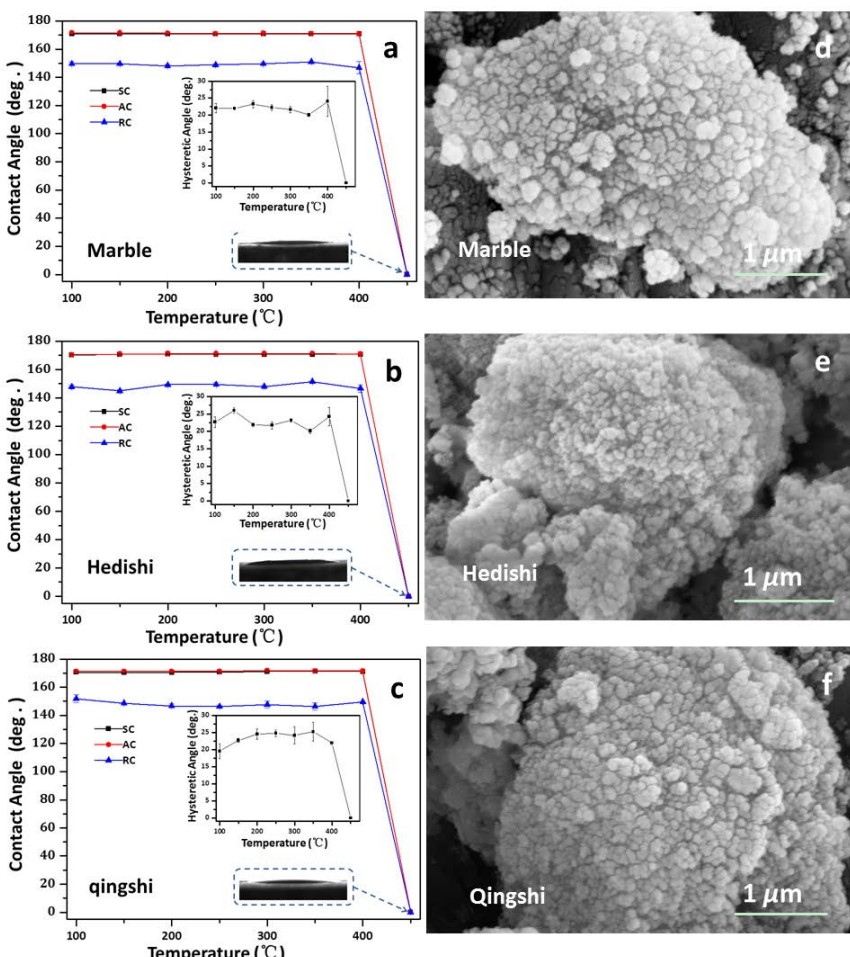

**Figure 6.** Variation of static, advancing, and receding water CAs with annealing temperature of the self-cleaning surfaces on marble (**a**), Hedishi (**b**), and Qingshi (**c**), and their SEM images (**d**–**f**) after annealing at 450 °C.

## 4. Discussion

### 4.1. Growth Mechanism of PDA Layer

The polymerization rate of the PDA is influenced by the concentration of dopamine, temperature, pH value, and concentration of $O_2$ in Tris-HCl solution [23–27]. In this work, the pH value of the Tris-HCl solution was fixed at 8.5, and the polymerization was conducted in air and at room temperature. As a result, the concentration of dopamine is the main factor that influences the polymerization. However, the polymerization rate will be saturated when the concentration of the dopamine reaches a certain value [28]. During the first 10 h, the thickness of the PDA increases relatively rapid, and then the increase becomes moderate. The PDA layer remains unchanged after 24 h of polymerization. The concentration of dopamine, the temperature, and the polymerization time can influence the roughness of the PDA surface [29,30].

### 4.2. Mechanism of the Liquid Wettability

Aiming to explain the impact of surface roughness on the apparent CA of liquid droplets on solid surfaces, Wenzel and Cassie developed two models, respectively. The Wenzel model suggests full contact at the liquid/solid interface in which surface roughness increases the actual contact area. As a consequence, the wettability of the solid surface is amplified, and the apparent CA can be expressed as follows [31–34]:

$$\cos\theta = r\cos\theta_0 \tag{1}$$

where $\theta$ is the apparent CA of droplets on the rough surface, and $r$ is the surface roughness factor ($r > 1$), which is the ratio of actual contact area of the solid surface with the liquid to the geometric area in contact with the droplet. In Equation (1), $\theta_0$ is the equilibrium CA of the droplet on an ideal smooth surface of the same material, which is given by Young's equation, $\cos\theta_0 = (\gamma_{sv} - \gamma_{sl})/\gamma_{lv}$, where $\gamma$ is the interfacial tension; and the subscripts $s$, $v$, and $l$ are attribute to the solid, vapor, and liquid phases, respectively.

The Cassie model suggests an incomplete contact at the liquid/solid interface in which air bubbles are trapped in micro-pockets formed on a hydrophobic rough surface, leading to a composite interface that gives the apparent CA as follows [35–37]:

$$\cos\theta = f(\cos\theta_0 + 1) - 1 \tag{2}$$

where $f$ is the ratio of the solid area in contact with the liquid to the geometric area in contact with the droplet ($f < 1$).

According to the Wenzel model, the micro-pockets at the solid–liquid interface will be filled with the liquid. As a consequence, the droplets can hardly slide, or the sliding angle is large on the surfaces. Hence, the Wenzel model cannot fulfill the sliding requirement of self-cleaning surfaces; that is, the sliding angle is less than 5° (or the HA is less than 20°). Based on the discussion above, from the sliding requirement point of view, constructing a Cassie state is suitable for self-cleaning surfaces. In order to achieve a Cassie state, both a hierarchical micro-nanometer scale roughness and low-surface-energy material are necessary.

When the $Al_2O_3$ emulsion and FPA solution were mixed, FPA monolayers were self-assembled on the surfaces of $Al_2O_3$ particles to form $Al_2O_3$@FPA microparticles (in the cases of nano-agglomerates) and nanoparticles. The FPA molecules were bonded with $Al_2O_3$ via P-O-Al covalent bonds. Water/oil repellency of $Al_2O_3$@FPA particles come from the $CF_3$ terminating groups and the -$CF_2$-$CF_2$- chains of the FPA molecules [32].

After being treated with PDA and $Al_2O_3$@FPA, the stone surfaces presented hierarchical structures, including micro- and nanoscale roughness (Figure 5). For the as-prepared self-cleaning stone surfaces, the Cassie state was constructed when they made contact with liquid droplets, and the interstices and holes on the stone surfaces acted as micro-pockets, which attract air at the interfaces of stone and liquid droplets. As a consequence, the actual

solid–liquid contact area is very small. In this work, the $f$ can be estimated to be ~2% according to Equation (2). The result agrees will with those reported in the literature [21,22].

### 4.3. Thermal Stability and Durability

As shown in Figure 6a–c, the CAs of the three treated stones remained constant (~170°) after they were annealed at temperatures up to 400 °C. This experimental result is a reflection that the self-cleaning functionality of the three treated stones is thermally stable up to that temperature. The thermal stability of surfaces is superior to those surfaces coated with fluorinated low-surface-tension chemicals [21,22,32]. Upon annealing at 450 °C, the CAs dropped to much less than 10°, suggesting that the surfaces of the three treated stones became highly hydrophilic. Photographs of a water drop wetting the surface at 450 °C is shown in the inserts in Figure 6a–c. The hydrophilicity of the surfaces annealed at 450 °C is readily explained by the oxidation [32] of the fluorocarbon chains of the FPA molecules in the SAMs attached to $Al_2O_3$ particles. In the SEM images shown in Figure 6d–f, one can see that the morphology of the surfaces is almost unchanged. This is because the FPA molecular chains are ~2 nm, so their oxidation will not impact the microstructures of the surface. The HAs shown as inserts in Figure 6a–c are 20–25° for samples annealed at temperatures up to 400 °C. When annealed at 450 °C, the HAs are close to zero, because both the advancing and receding angles are close to zero, due to the fact that the surface is hydrophilic now.

The self-cleaning stone surfaces were exposed to the air for more than 180 days, and water/oil CAs measurement were carried out. The wettability of the surfaces remained unchanged, thus indicating that the durability of the protective layers is excellent and is comparable with the work conducted by our group [20]. We believe that the lifetime of the protective coatings is over one year.

### 5. Conclusions

Dopamine layers polymerized on polished natural marble, Hedishi, and Qingshi surfaces with inherent microscale interstices and holes served as an adhesive that was highly hydrophilic. Alumina ($Al_2O_3$) nanoparticles derivatized by self-assembled monolayers of a fluorophosphonic acid were strongly attached to the polymerized dopamine surface, rendering the surface superhydrophobic and superoleophobic, as evidenced by a static water contact angle of ~170° and a static peanut-oil contact angle of 145°. This self-cleaning stone surface was stable up to 400 °C, beyond which the fluorophosphonic acid chains were oxidized, leaving the surface highly hydrophilic. This work developed an effective process for making a self-cleaning surface for natural stone that can be scaled up for industrial applications.

**Author Contributions:** Data curation, Z.D. and Z.X.; investigation, X.Y.; methodology, Z.D. and Z.X.; supervision, Y.H.; validation, H.-Y.N. and Y.H.; visualization, B.Z.; writing—original draft, Z.D. and Z.X.; writing—review and editing, H.-Y.N. and Y.H. All authors have read and agreed to the published version of the manuscript.

**Funding:** This work was funded by National Natural Science Foundation of China (NSFC) grant number 11764003 and Yunnan Provincial Department of Education Research Fund grant numbers 018JS414 and 2020J0544.

**Institutional Review Board Statement:** Not applicable.

**Informed Consent Statement:** Not applicable.

**Data Availability Statement:** The datasets used and analyzed during the current study are available from the corresponding author upon reasonable request.

**Conflicts of Interest:** The authors declare no conflict of interest.

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
