# Peer review of "Natural Stones with a Self-Cleaning Surface via Self-Assembled Monolayers"

_applsci, doi:10.3390/app12094771_

Round 1

Reviewer 1 Report

The manuscript reports the Natural stones with a self-cleaning surface via self-assembled monolayers. The experiments were well designed and the analysis was clear and insightful. Also, the topic of this paper is likely within the scope of applied science, which may make potential contribution to the related research field as well. Thus, the reviewer considered it worthwhile publishing in applied science. There are some concerns needed to be addressed before publication.

Comments:

  1. The abbreviations should be defined in the first appearance.
  2. Remove " Superhydrophobicity " from the keywords list.
  3. The use and purity of chemicals used for the fabrication of films should be included in a separate section.
  4. How did you perform thermal stability for the as-prepared self-cleaning surfaces? Needs some clarification in the experimental part.
  5. More modern surface analyses such as AFM and EDAX should be included.
  6. Section liquid wettability Mechanism needs more explanation.
  7. From the data obtained it is clear the self-cleaning surface for natural stones results obtained are satisfactory, it would be very convenient to compare the results with a well-known systems.
  8. What about the film thickness?. Should be included and how it is measured.
  9. More explanation/clarification/significance is needed for Fig 6
  10. There are some typos in the manuscript. Please correct them in the revised manuscript.

Reviewer 2 Report

The work presented here describes a method to develop self-cleaning property in different types of natural stones. Authors have approached the problem by employing polydopamine (PDA) as the adhesive layer on the stone surface, followed by depositing  Al2O3 nanoparticles derivatized by self-assembled monolayers of a fluorophosphonic acid (FPA). The work is interesting and the technique proposed / results are promising. Therefore, I recommend the acceptance of the work in its present form with minor modifications as suggested below. 

  1. The work presented here may be applicable to protect the heritage buildings as described by the authors. However, what is the life time of the protective coating ? What is the efficiency of self cleaning property of the coating with time? This should be addressed.
  2. Surface roughness is one of the property authors discuss to attribute the self cleaning property based on contact angle. However, what method is used to evaluate the surface roughness? Did authors perform experiments like AFM or surface roughness measurements? If so provide the data within the manuscript. 

Round 2

Reviewer 1 Report

All comments have been addressed, the manuscript could be accepted